# Lung Subregion Partitioning by Incremental Dose Intervals Improves Omics-Based Prediction for Acute Radiation Pneumonitis in Non-Small-Cell Lung Cancer Patients

**DOI:** 10.3390/cancers14194889

**Published:** 2022-10-06

**Authors:** Bing Li, Xiaoli Zheng, Jiang Zhang, Saikit Lam, Wei Guo, Yunhan Wang, Sunan Cui, Xinzhi Teng, Yuanpeng Zhang, Zongrui Ma, Ta Zhou, Zhaoyang Lou, Lingguang Meng, Hong Ge, Jing Cai

**Affiliations:** 1Department of Health Technology and Informatics, The Hong Kong Polytechnic University, Hong Kong, China; 2Department of Radiation Oncology, Affiliated Cancer Hospital of Zhengzhou University & Henan Cancer Hospital, Zhengzhou 450008, China; 3Department of Radiation Oncology, Stanford University School of Medicine, Palo Alto, Stanford, CA 94305, USA

**Keywords:** dosiomics, multi-omics, radiation pneumonitis, radiomics, radiotherapy

## Abstract

**Simple Summary:**

Acute radiation pneumonitis (ARP) is one of the common radiation toxicities in patients with non-small-cell lung cancer (NSCLC) treated by radiotherapy. The prediction for ARP can benefit clinical management and thereby improve the quality of life of patients. Markers derived from radiomics and dosiomics features can be adopted to predict ARP. Moreover, the features from lung subregions can improve the prediction for ARP compared to the whole lung region. The previous studies using lung subregion features for predicting ARP ignored the low-dose covered lung region. In this manuscript, we proposed an incremental-dose-interval-based lung subregion segmentation method by covering the low-dose lung region for ARP modeling and evaluated its effectiveness in improving ARP predictions using pre-treatment CT radiomics features and planning dose features for patients with NSCLC treated by radiotherapy. The performance among ARP models using features from our proposed lung subregion and the whole lung were compared for both multi-omics and single-omics. The results could provide insight into the subregion’s ability to predict the ARP and could be useful in predicting the risk of radiation-induced toxicity. This manuscript and its results were neither published in any conferences nor journals.

**Abstract:**

Purpose: To evaluate the effectiveness of features obtained from our proposed incremental-dose-interval-based lung subregion segmentation (IDLSS) for predicting grade ≥ 2 acute radiation pneumonitis (ARP) in lung cancer patients upon intensity-modulated radiotherapy (IMRT). (1) Materials and Methods: A total of 126 non-small-cell lung cancer patients treated with IMRT were retrospectively analyzed. Five lung subregions (SRs) were generated by the intersection of the whole lung (WL) and five sub-regions receiving incremental dose intervals. A total of 4610 radiomics features (RF) from pre-treatment planning computed tomographic (CT) and 213 dosiomics features (DF) were extracted. Six feature groups, including WL-RF, WL-DF, SR-RF, SR-DF, and the combined feature sets of WL-RDF and SR-RDF, were generated. Features were selected by using a variance threshold, followed by a Student *t*-test. Pearson’s correlation test was applied to remove redundant features. Subsequently, Ridge regression was adopted to develop six models for ARP using the six feature groups. Thirty iterations of resampling were implemented to assess overall model performance by using the area under the Receiver-Operating-Characteristic curve (AUC), accuracy, precision, recall, and F1-score. (2) Results: The SR-RDF model achieved the best classification performance and provided significantly better predictability than the WL-RDF model in training cohort (Average AUC: 0.98 ± 0.01 vs. 0.90 ± 0.02, *p* < 0.001) and testing cohort (Average AUC: 0.88 ± 0.05 vs. 0.80 ± 0.04, *p* < 0.001). Similarly, predictability of the SR-DF model was significantly stronger than that of the WL-DF model in training cohort (Average AUC: 0.88 ± 0.03 vs. 0.70 ± 0.030, *p* < 0.001) and in testing cohort (Average AUC: 0.74 ± 0.08 vs. 0.65 ± 0.06, *p* < 0.001). By contrast, the SR-RF model significantly outperformed the WL-RF model only in the training set (Average AUC: 0.93 ± 0.02 vs. 0.85 ± 0.03, *p* < 0.001), but not in the testing set (Average AUC: 0.79 ± 0.05 vs. 0.77 ± 0.07, *p* = 0.13). (3) Conclusions: Our results demonstrated that the IDLSS method improved model performance for classifying ARP with grade ≥ 2 when using dosiomics or combined radiomics-dosiomics features.

## 1. Introduction

Lung cancer is one of the most common cancer malignancies, ranking second place in the incidence rate and first in mortality rate worldwide [1]. Radiotherapy (RT) remains a standard of care for advanced or non-operable lung cancer patients [2,3,4,5,6]. Acute radiation pneumonitis (ARP) is one of the most debilitating complications occurring within 6 months after RT commencement. Severe (grade ≥ 2) ARP can significantly degrade patients’ quality of life, cause treatment interruption, and thereby hinder therapeutic effect and worsen prognosis. Therefore, pre-treatment identification of high-risk patients with severe ARP is essential for effective clinical management of lung cancer patients by providing timely interventions.

Over the past decades, there have been extensive studies conducted with the aim of identifying potential ARP risk factors, among which radiation dose has long been considered the major attribute for inducing ARP in lung cancer patients [7,8,9,10,11,12]. Tremendous efforts have been made to investigate the role of dosimetric factors derived from one-dimensional dose-volume histograms (DVHs) in ARP occurrence. For example, Boonyawan et al. [8] reported an elevated risk of ARP when DVH parameters of V10 and V20 increased. However, their predictive power is considered limited due to the lack of information on spatial dose distributions within the entire lung volume, which impedes a more thorough understanding of the degree of dose heterogeneity deposited to particular sub-regions of the lung.

More recently, the advent of Radiomics, enabling the calculation of high-throughput imaging features from the entire tissue volume for cancer predictive modeling, has motivated scientists to derive three-dimensional spatial dose distribution information from dose-map images, referred to as Dosiomics. There exist several studies on applying Dosiomics features for ARP predictions in the body of literature. For instance, Liang et al. [9] investigated the correlation between ARP and dosiomics features extracted from the entire lung volume and demonstrated the superiority of dosiomics over the traditional DVH parameters and normal tissue complication possibility (NTCP) factors. Furthermore, radiomics features have been combined with DVH factors in an attempt to achieve better predictive performance for ARP. Jiang et al. [11] combined DVH parameters and radiomics features extracted from pre-treatment planning computed tomography (CT) images, and they reported that the joint radiomics-DVH model yielded an improved prediction for ARP. Nevertheless, these studies have placed little emphasis on the intra-pulmonary heterogeneity in biologic response upon treatment perturbation.

In view of this, a few recent studies have been carried out on lung subregion-based analysis for ARP prediction. Generally, lung segmentations in the prediction of ARP can be divided into three categories, such as anatomy-based [13], lung function-based [14,15], and dose-based [16,17,18]. The dose-based lung segmentation is widely investigated as considering the dose information into the radiation-induced toxicity. For example, Chopra et al. [16] investigated the predictability of radiomics and dosiomics features extracted from two lung sub-regions that received doses greater than 5 Gy and 20 Gy, respectively. They demonstrated an improved ARP classification using features from lung subregions than those from the whole lung. Similarly, Adachi et al. [17] reported an improvement in ARP prediction using dosiomics features from lung subregions receiving doses greater than several dose thresholds compared with those from the whole lung. Notwithstanding, the existing dose-thresholding methods used in lung subregion partitioning have placed little attention on the low-dose lung region, where its association with ARP risk remains unclear.

Therefore, we proposed an Incremental-Dose-interval-based Lung Subregion Segmentation (IDLSS) approach for ARP modeling in this study. Specifically, the whole lung was partitioned into five non-overlapped subregions based on the intersection of the whole lung and five specified dose intervals that cover an incremental range of prescription dose to the entire lung volume. In this work, we aimed to (1) investigate the effectiveness of the proposed IDLSS method for predicting ARP with grade ≥ 2 in non-small-cell lung cancer (NSCLC) patients following IMRT treatment and (2) analyze the responsiveness of each omics feature to the whole lung region and lung subregions generating from IDLSS approach, which may provide a new prospective approach for radiomics related method.

To this end, we developed six ARP models based on the CT-based radiomics and/or dosiomics features extracted from both the whole lung and lung subregions and evaluated the model performance using a set of evaluation metrics. The success of this study would provide physicians with a more thorough understanding of the contributions of different lung SRs at various dose levels in the occurrence of ARP events in NSCLC patients following IMRT, potentially assisting physicians in prescribing timely interventions for at-risk patients.

## 2. Materials and Methods

### 2.1. Study Workflow

The overall workflow of this study contains five steps, as shown in Figure 1.

### 2.2. Patient Data

The study was approved by the ethical committee of the Affiliated Cancer Hospital of Zhengzhou University. We retrospectively collected clinical and RT data of (stage III A-B) non-small-cell lung cancer patients treated between 2015 and 2019. All patients received curative (chemo-) radiotherapy with a prescribed dose of 50–72 Gy and a fractional dose of 1.8–2.0 Gy (5 days/week) using 6 MV IMRT technique. Patients who received chest RT before the lung cancer treatment were excluded. Radiation pneumonitis was graded following the Common Terminology Criteria for Adverse Events (CTCAE) V4.0 by a radiation oncologist with more than 5 years of experience. In this study, the occurrence of severe (grade ≥ 2) ARP that happened within 6 months post RT commencement was chosen as the clinical endpoint.

The pre-treatment planning CT images, planning dose distributions, and lung segmentations were retrieved from the treatment planning system. All CT images were scanned on the Brilliance Big Bore CT scanner (Philips Electronics, Eindhoven, The Netherlands) with a slice thickness of 3 mm. The dose grid size was set as 3 mm for dose calculation, and the lung volumes were contoured by a radiation oncologist with more than 5 years of experience.

### 2.3. Lung Sub-Region Partitioning

Lung subregions (SRs) were generated based on the proposed IDLSS method, where 5 equally spaced dose intervals covered 5 incremental ranges of the prescription dose to the whole lung volume. For example, as illustrated in Figure 2a, the lung SR with a dosing interval of 10–20 Gy (enclosed by light green region) is defined as the intersection of the whole lung (enclosed by light orange region) and the volume receiving dose ranging between 10 and 20 Gy (enclosed by light blue region). Similarly, the other 4 lung SRs were generated for dose intervals of 0–10 Gy, 20–30 Gy, 30–40 Gy, and 40–50 Gy, as illustrated in Figure 2b. The incremental dose interval of 10 Gy was chosen based on the following two considerations: (1) the volume of an SR should be large to contain a sufficient number of image voxels for characterizing texture information, such as gray level heterogeneity; (2) more SRs may provide a more thorough understanding on intra-pulmonary heterogeneity in radiosensitivity at a variety of dose levels.

### 2.4. Feature Extraction

In the study, 4610 radiomics features were calculated from the pre-treatment planning CT image for each lung SR and the whole lung by using a Python package Pyradiomics [18]. Laplacian-of-Gaussian (LoG) filters with a sigma value of 1, 3, and 6 mm and coil f1 wavelet filters with combinations of high (H) and low (L) pass filtering along three imaging axes (LLL, HLL, LHL, LLH, LHH, HLH, HHL, and HHH) were applied to filter the original image for generating additional sets of radiomics features. All the images were discretized into fixed bin counts by 5 different bin count numbers (20, 50, 100, 150, and 200) prior to feature extraction, resulting in a total of 27,660 radiomics features.

The dose features included in this work were commonly used in previous studies [9,19,20,21]. They can be categorized into scale-invariant 3D dose moments, DVH parameters, and dosiomics features.
Scale-invariant 3D dose moments [19]: They describe the weighted dose center within the organ-at-risk (OAR) volume with varying orders along with anterior–posterior, medial–lateral, and craniocaudal directions [22]. In this study, the maximum order of 3 was chosen for each dimension, resulting in 64 possible combinations of orders. Scale invariance can be calculated by: ηpqr = μpqrμ000p+q+ r 3+1, where p, q, and r are the orders in three directions, and $\mu_{pqr}$ is central moments which are defined in [19]. Since the order of p=0, q=0, r=0 results in a constant value of 1, a total of 63 dose moments were included in the dosiomics feature set.DVH parameters [20,21]: DVH summarizes the dose accumulation within a volume. It is defined as the isodose volume at varying levels of doses and is widely used in the clinic for convenient dose comparisons. DVH parameters, which are the dose values at specific volumes or volume values at specific doses, were commonly used as the evaluation metrics for plan quality assessment. In this study, we selected multiple DVH parameters of Vx and Dx from the DVH curve, where Vx was the volumes or relative volumes (of the whole organ) receiving more than x Gy, and Dx was the dose (Gy) to x% of the whole lung.Dosiomics [9]: A total of 91 first-order and higher-order radiomics features were extracted from the dose map to describe the dose histogram statistics and dose texture. Only the original dose map was employed without further preprocessing.In total, 213 dose features for each lung SR and the whole lung were extracted, resulting in 1278 dose features in total.

### 2.5. Feature Selection

As shown in Figure 3b, feature selection was performed in an iterative process with five steps for each feature set. (1) At each iteration, 70% of the balanced data (88 cases) were subsampled independently from the whole data set. (2) For the sampled data, features with zero variances were filtered out, and the remaining features were standardized independently by removing the mean and scaling to unit variance (0–1) to avoid bias from data skew and scale [23]. (3) The Student’s T-test was subsequently adopted to assess associations between the standardized features and the clinical outcome (ARP). Features with a *p*-value larger than 0.1 were further eliminated. (4) The previous iteration (steps 1–3) was repeated over 100 times to obtain 100 feature sets. The most (10% of total feature number or at least top 10 features) frequently occurring features were kept by considering the 100 feature sets and proceeded with a feature redundancy test. (5) By calculating the Pearson correlation coefficient (R) between each pair of features, the feature with a higher mean correlation with the rest of the features in each highly correlated feature pair (R > 0.5 [24]) was removed. Eventually, the remnant features were used for model construction.

### 2.6. Model Construction and Evaluation

The flow chart of the model construction procedure used in this study is summarized in Figure 3a. The procedure contains three steps: (1) the data were first divided into a training set (70%) and a testing set (30%) with a stratified sampling approach that maintained the same event distribution in both datasets; (2) model development was performed by using the Ridge algorithm on the training set with 5-fold cross-validation; (3) lastly, model performance were evaluated in both datasets using multiple evaluation metrics including area under the ROC curve (AUC), accuracy, precision, recall, and f1-score. A total of 30 train-test splits were used during model evaluation to assess the average model performance under sampling variations [25]. Student’s *t*-test was employed for model comparison.

In this study, six ARP classification models were generated. They included two classification models that contain a combination of radiomics and dose features extracted from the whole lung (WL-RDF) and the lung subregions (SR-RDF); and another four classification models that contain radiomics-alone or dose-alone features extracted from the whole lung (WL-RF, WL-DF) and the lung subregions (SR-RF, SR-DF). All the models were developed by using the analysis module Scikitlearn in Python [26].

## 3. Results

### 3.1. Patient Characteristics

A total of 126 (stage III A-B) non-small-cell lung cancer patients treated between 2015 and 2019 were considered eligible for inclusion in this study. Patient characteristics are summarized in Table 1. The average age of patients was 61 years with a standard deviation (STD) of 8.8 years, which belongs to the lower age group compared to other reports and clinical practice [27]. As shown in Table 1, approximately two-thirds of the patients were affected by squamous carcinoma cancer. Additionally, the majority of patients have received sequential chemoradiotherapy.

### 3.2. Selected Features

For the SR-RDF model, a total of 33 features (Radiomics:22, Dosiomics: 11) were selected. Of note, more than 50% of the selected dosiomics features came from lung sub-regions receiving less than 20 Gy (0–10 Gy: 3, 10–20 Gy: 3, 20–30 Gy: 1, 30–40 Gy: 1, 40–50 Gy: 3); and approximately 20% for the selected radiomics features originated from lung sub-regions receiving less than 20 Gy (10–20 Gy: 4, 20–30 Gy: 4, 30–40 Gy: 8, 40–50 Gy: 6). For the WL-RDF model, a total of 31 features (Radiomics: 28, Dosiomics: 3) were selected. The three dosiomics were glrlm_RunEntropy, firstorder_Energy, and scale-invariant dose moment dose_moment_2_3_3. For the other four models, the total numbers of selected features were 4 for WL-RF, 16 for WL-DF, 17 for SR-RF, and 22 for the SR-DF model. The names of all the selected features for the 6 developed models can be found in Appendix A.

### 3.3. Model Evaluation

Predictive performances of the 6 developed models are summarized in Table 2, and the corresponding STDs are listed in Appendix A. Additionally, the ROC curves of each model in both train and test sets are plotted in Appendix A.

For the joint radiomics-dosiomics models, as shown in Table 2, the SR-RDF model achieved the best classification performance for predicting ARP among the six developed models, yielding an average (±STD) AUC of 0.98 ± 0.01/0.88 ± 0.05 in the training and testing set, respectively; while the WL-RDF model performed significantly worse than the SR-RDF model with the average (±STD) AUC of 0.90 ± 0.02/0.80 ± 0.04 (*p* < 0.001) in the training and testing set, respectively.

For the dose-alone models, as demonstrated in Figure 4, the SR-DF model significantly outperformed the WL-DF models in both the training set (average (±STD) AUC: 0.88 ± 0.03 vs. 0.70 ± 0.03, *p* < 0.001) and testing set (average (±STD) AUC: 0.74 ± 0.08 vs. 0.65 ± 0.06, *p* < 0.001).

For the radiomics-alone model, however, the SR-RF model performed significantly better than the WL-RF model only in the training set (average (±STD) AUC: 0.93 ± 0.02 vs. 0.85 ± 0.03, *p* < 0.001), but not in the testing set (average (±STD) AUC: 0.79 ± 0.05 vs. 0.77 ± 0.07, *p* < 0.13).

## 4. Discussion

In this study, we proposed an Incremental-Dose-interval-based Lung Subregion Segmentation (IDLSS) method and evaluated its effectiveness in ARP predictions in non-small-cell lung cancer (NSCLC) patients following IMRT. CT-based radiomics-alone (RF), dosiomics-alone (DF), and combined Radiomics-Dosiomics features (RDF) extracted from whole-lung (WL), and IDLSS-segmented lung sub-regions (SRs) were employed for developing the six ARP prediction models, which were then analyzed and compared in aspects of model predictability. Results of this study would provide physicians with a more thorough understanding of the contributions of different lung SRs at various dose levels in the occurrence of ARP events in NSCLC patients, potentially assisting physicians in prescribing timely interventions for at-risk patients.

Results of this study demonstrated that the SR-RDF model achieved the highest scores in all the evaluating metrics (AUC: 0.88, Accuracy: 0.83, Precision: 0.69, Recall: 0.79, and F1-score: 0.73) in the testing sets among all the studied models (Table 2). Particularly, it performed significantly better than the WL-RDF model in terms of AUC in the training set (average (±STD) AUC: 0.98 ± 0.01 vs. 0.90 ± 0.02, *p* < 0.001) and testing set (average (±STD) AUC: 0.88 ± 0.05 vs. 0.80 ± 0.04, *p* < 0.001), as shown in Table 2 and Figure 4. This outstanding performance implicates that the IDLSS-partitioned lung SRs may offer greater sensitivity of the combined Radiomics-Dosiomics features for ARP prediction than the whole lung.

Among the selected features in the SR-RDF model, approximately one-third (10 out of 33), including more than 50% (6 out of 11) of the selected DF and around 20% (4 out of 22) of the selected RF, originated from low-dose regions of lung SRs (0–20 Gy) (Appendix A), suggesting its unneglectable role in contributing to ARP in NSCLC patients. The finding that over 50% of the selected dosiomics features originate from low-dose subregions (<20 Gy) could be potentially ascribed to the larger inter-patient discrepancy in dose heterogeneity in low-dose regions than the high-dose regions. In IMRT treatment planning of lung cancer, highly conformal and homogeneous radiation dose deposition to the tumors (i.e., the high-dose subregion) is typically required for tumor control; this may likely lead to the phenomenon that the dose homogeneity within the high-dose subregions is less dissimilar between patients and, hence, might not be sufficiently sensitive in predicting ARP. While in low-dose subregions, the inter-patient disparity in the dose homogeneity pattern of lung tissue is larger, which might, to a degree, explain the larger proportion of selected dosiomics features in these areas. In addition, approximately 50% (12 out of 22) of the selected RF stemmed from high-dose regions of lung SRs (30–50 Gy) (Appendix A), which suggests a correlation between severe ARP and radiomics features in the high-dose lung volumes. We speculated that these RFs may quantify lung function information such as perfusion or ventilation, which are more heterogeneous in high-dose regions near tumor. Such a correlation between lung function and dose levels was implied in the study by Yvette et al. [14], where the perfusion-weighted mean regional dose was found to be lower than the non-weighted one in the irradiated area. On the other hand, based on our speculation, the dominating high functional lung tissue in the low-dose subregions could be the reason for more selected dose features. Several studies have demonstrated the high predictive power of dosimetric factors under the high lung function region [15,28,29,30,31]. For example, Shanon et al. discovered that V20 Gy, V5 Gy, and mean dose in the high ventilation area have significant correlations with radiation pneumonitis, while no such pattern was found for the low function region [32]. Nevertheless, the relationship between RFs and lung functions remains unclear; and further investigations on the sensitivity of lung function to ARP at various dose levels are warranted in the future.

On the contrary, the WL-RDF model contained remarkably less amount to dosiomics features compared to the SR-RDF model (3 vs. 11), which may highlight the higher sensitivity of regional dose distribution over the global-scale dose information for ARP prediction. This interpretation can be reflected by both the superiority of the SR-RDF model over the WL-RDF model (average (±STD) AUC: 0.88 ± 0.05 vs. 0.80 ± 0.04, *p* < 0.001) and the SR-DF model over WL-DF model (average AUC: 0.74 vs. 0.65, *p* < 0.001) in the testing set in this study. On the other hand, it is worth noting that only three dosiomics (one first-order, one texture dosiomics feature, and one scale-invariant dose moment) were included in the WL-RDF model (Supp. Table 1). None of the previously reported DVH parameters [33,34,35], such as V20 and mean dose of the whole lung, were selected in the present work, suggesting that the spatial texture dose features representing the 3D dose distribution may be more sensitive for the prediction of ARP. This finding is consistent with a previous study [10], where five dosiomics features from the Grey-Level Co-Occurrence Matrix achieved the best prediction results for ARP.

Intriguingly, we found that the SR-RF model performed significantly better than the WL-RF model only in the training set (average (±STD) AUC: 0.93 ± 0.02 vs. 0.85 ± 0.03, *p* < 0.001), but not in the testing set (average (±STD) AUC: 0.79 ± 0.05 vs. 0.77 ± 0.07, *p* = 0.13) (Table 2 and Figure 4). This could be in part due to the deficiency of the proposed IDLSS approach in enhancing the sensitivity of RF from lung SRs for ARP prediction in this work, as compared to that from the whole lung. However, the underlying reason is still unknown, and a close scrutinization in this regard should be carried out in the future.

On the other hand, it was observed that the RF consistently outperformed DF in both whole lung-based and lung subregion-based models. For instance, the SR-RF model performed significantly better than the SR-DF model in the testing set (average (±STD) AUC: 0.79 ± 0.05 vs. 0.74 ± 0.08, *p* < 0.001). Similarly, the WL-RF model yielded significantly higher predictability in the testing test (average (±STD) AUC: 0.77 ± 0.07 vs. 0.65 ± 0.06, *p* < 0.01). This finding may probably be ascribed to the superiority of radiomics over dosiomics in assessing intrinsic radiation-induced tissue response. Notwithstanding, it is worth pointing out that the combined Radiomics-Dosiomics features (i.e., the RDF) consistently outperformed RF and DF in both whole lung-based (*p* < 0.05) and lung subregion-based (*p* < 0.001) models. This observation could be attributed to the complementary role between DF and RF in the identification of modifications underlying ARP occurrence. Similarly, one previous study reported superior severe radiation pneumonitis risk estimates using functional dose-volume parameters from the combination of single-photon emission CT images and the dose distribution compared with standard parameters [36].

Last but not least, the incidence of ARP in the present work was found to be 50.8%, which is higher than the range from 16.9% to 45.1% (Table 3) reported by previous studies [11,16,17,37,38,39,40,41]. This could be due to the inclusion of only stage III A-B patients who required a higher overall dose to the whole lung in order to provide sufficient dose coverage to the large-sized tumor, hence resulting in an increased incidence rate of severe ARP. The previous severe ARP modeling studies reported a wide range of performance, with the testing AUCs ranging from 0.68 to 0.94, as shown in Appendix A. Apart from one study [10], the severe ARP prediction models generated by our proposed IDLSS method (SR-RDF) had a higher AUC value than other studies in testing AUCs. However, one study, which reported a higher model performance than the results of this work, used only one data split of a small-sized cohort (*n* = 79) to generate a training and a testing set, which greatly reduced the significance of the performance assessment results [10].

There exist several limitations in our study. First, only single institutional retrospective data were used to train the model and assess its predictability. A large multicenter cohort should be further investigated to increase the credibility of our conclusions. Second, the proposed IDLSS approach considers an incremental dose interval of 10 Gy, which may not be optimal in the context of ARP onset in NSCLC patients. A further study using varying sizes of dose intervals should be conducted to obtain a more thorough understanding of contributions of different lung SRs at different dose levels for ARP prediction. Third, reproducibility of the radiomics and dosiomics features were not considered in our study. Zwanenbur et al. [42] have demonstrated that certain radiomics features from the gross tumor volume have limited reproducibility under perturbations. As we know, the main limit to the actual clinical implementation of radiomics is reproducibility [43]. Hence, further investigation of the robustness of the selected features in the prediction model is warranted.

## 5. Conclusions

We proposed a new IDLSS method for predicting ARP of grade ≥ 2 in lung cancer patients following IMRT treatment. It was found that the IDLSS method improved model performance for classifying the ARP when using dosiomics or combined radiomics-dosiomics features. Results of this study would provide physicians with a more thorough understanding of the contributions of different lung SRs at various dose levels in occurrence of ARP event in NSCLC patients, potentially assisting physicians in prescribing timely interventions for at-risk patients.

## Figures and Tables

**Figure 1 cancers-14-04889-f001:**
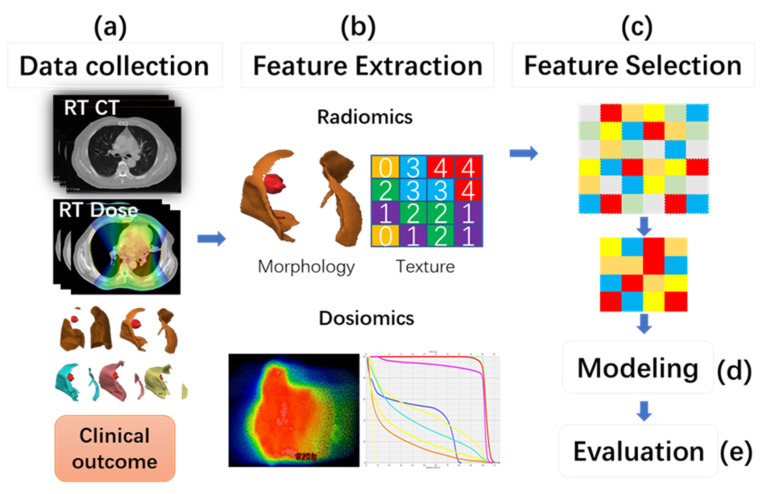
The overall workflow of the model construction. (**a**) collection of the clinical records and RT data, including pre-treatment CT images, planning dose distributions, and lung contours; (**b**) extraction of radiomics features (RF) and dosiomics features (DF), including DVH parameters using the planning CT image and RT dose distribution; (**c**) feature selection; (**d**) model construction; (**e**) model performance evaluation.

**Figure 2 cancers-14-04889-f002:**
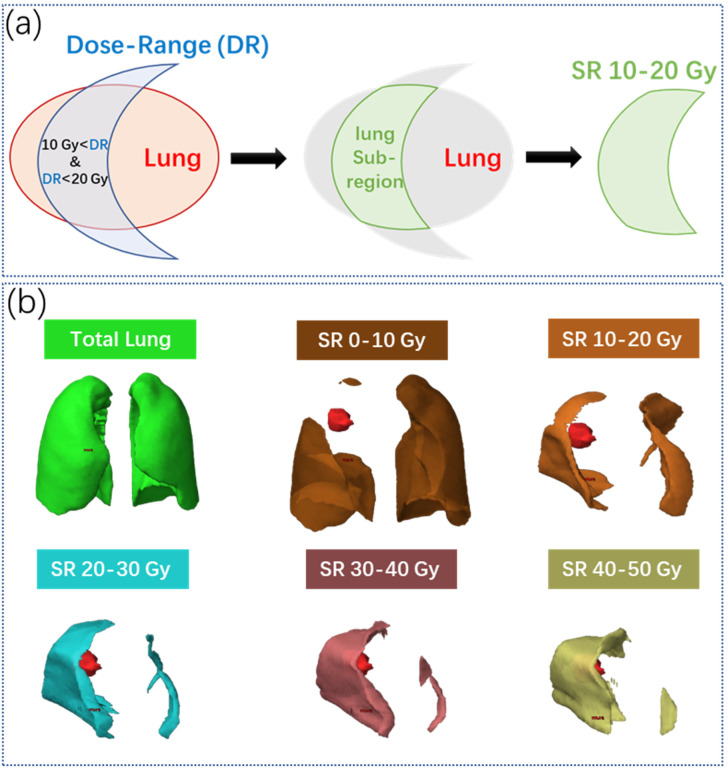
(**a**) An illustration lung subregion segmentation of 10–20 Gy. (**b**) The whole lung and the five lung subregions with dose intervals of 0–10 Gy, 20–30 Gy, 30–40 Gy, and 40–50 Gy of one example patient.

**Figure 3 cancers-14-04889-f003:**
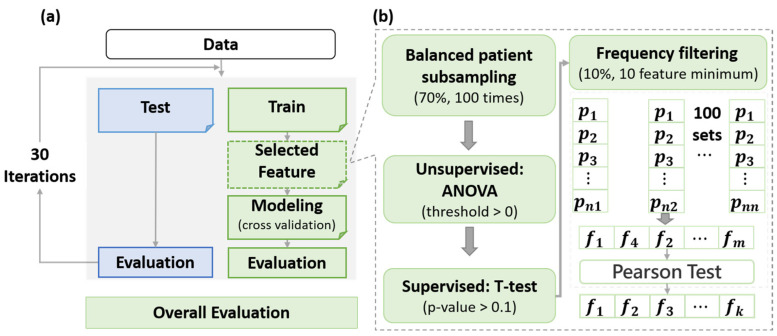
The flowchart of (**a**) classification model construction and (**b**) feature selection.

**Figure 4 cancers-14-04889-f004:**
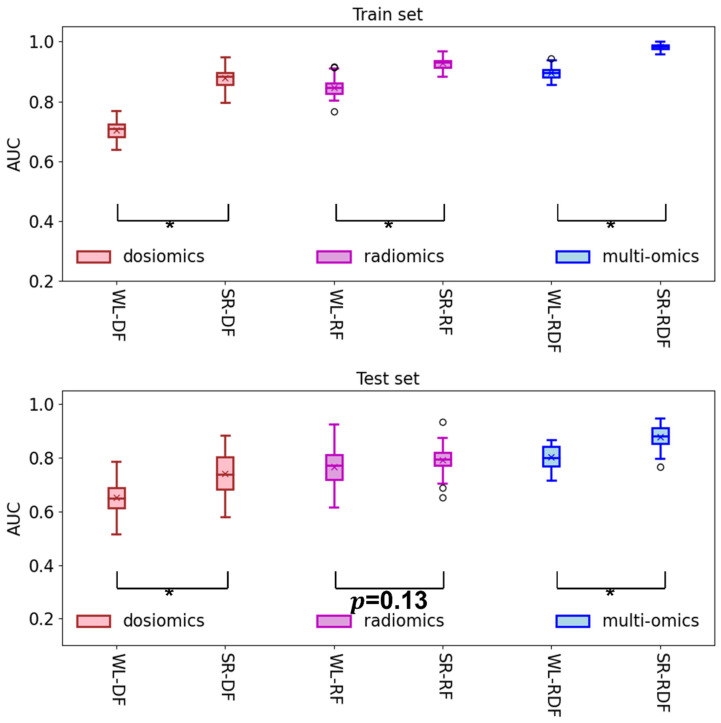
The model performance comparison between the models using subregion features and the whole lung features, i.e., SR-DF vs. WL-DF, SR-RF vs. WL-RF, and SR-RDF vs. WL-RDF, in the train and test AUCs. * *p* < 0.001.

**Table 1 cancers-14-04889-t001:** Patient characteristics.

Characteristics	Overall (126)
Gender	*p* = 0.04
Male (N/%)	109/86.5%
Female (N/%)	17/13.5%
Age, median (range)	61 (29–82) (*p* = 0.67)
Pathology	*p* = 0.46
SCC (N/%)	79/62.7%
ADC (N/%)	42/33.3%
Others (N/%)	5/4.0%
RT Dose, median (range)	60 (50–70) Gy (*p* = 0.94)
Smoking	*p* = 0.23
Activity or former (N/%)	97/77.0%
Never (N/%)	29/23.0%
Overall Stage	*p* = 0.30
IIIA (N/%)	80/63.5%
IIIB (N/%)	46/36.5%
Treatment method	*p* = 0.97
SCRT (N/%)	83/65.9%
CCRT (N/%)	42/33.3%
RT (N/%)	1/0.8%
ARP (N/%)	64/50.8%

Abbreviations: SCC: Squamous carcinoma cancer, ADC: Adenocarcinoma cancer, SCRT: Sequence chemoradiotherapy, CCRT: Concomitant chemoradiotherapy.

**Table 2 cancers-14-04889-t002:** The average evaluation results of the models using features of WL-DF, WL-RF, WL-RDF, SR-DF, SR-RF, and SR-RDF.

	Cohort	WL-DF	WL-RF	WL-RDF	SR-DF	SR-RF	SR-RDF
**AUC**	Train	0.70	0.85	0.90	0.88	0.93	0.98
	Test	0.65	0.77	0.80	0.74	0.79	0.88
**Acc**	Train	0.63	0.75	0.83	0.78	0.86	0.93
	Test	0.59	0.70	0.74	0.71	0.74	0.83
**Pre**	Train	0.41	0.56	0.67	0.59	0.71	0.82
	Test	0.38	0.49	0.55	0.51	0.54	0.69
**Re**	Train	0.68	0.75	0.81	0.81	0.86	0.96
	Test	0.63	0.69	0.66	0.66	0.65	0.79
**F1**	Train	0.51	0.64	0.73	0.68	0.78	0.88
	Test	0.47	0.56	0.59	0.57	0.59	0.73

Abbreviation: Acc: Accuracy; Pre: Precision; Re: Recall; F1: F1-score.

**Table 3 cancers-14-04889-t003:** The previous studies for predicting the ARP using single or multiple omics features for lung cancer patients treated with RT.

Reference	Features (*n*)	Classification	Methods	AUC	Patient Information
[37]	Radiomics (9)	ARP grade ≥ 2	Logistics regression	0.75	SBRT for 40 stages I NSCLC patients
[38]	Radiomics (8), DDF (5)	ARP grade ≥3	LASSO	0.68	IMRT/3DCRT for 192 NSCLC patients
[39]	DDF (5), Clinical factors (13), Cytokines (30), miRNAs (62), SNPs (60)	ARP grade ≥ 2	RF, SVM, MLP	0.831	RT for 106 NSCLC patients
[40]	DDF (11), Clinical factors (21)	ARP grade ≥ 2	RF	0.66	RT for 203 stage II–III NSCLC patients
[11]	Radiomics (TL-GTV) Multi-ROIs radiomics	ARP grade ≥ 2	SVM	0.71 0.94	VMAT for 79 stages I-IV lung cancer patients
[16]	Radiomics, Dosiomics, Clinical factors	ARP grade ≥ 2	RF	0.771 (V_20_) 0.763 (V_5_)	RT for 701 NSCLC patients
[41]	Radiomics (486)	ARP grade ≥ 2	Logistic regression	0.871 (Training) 0.756 (Testing)	SBRT For 275 stage I NSCLC patients
[17]	Dosiomics	ARP grade ≥ 2	LightGBM	0.846	SBRT for 685 NSCLC patients

## Data Availability

The datasets generated for this study are available on request to the corresponding author.

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
