# Peer review of "Lung Subregion Partitioning by Incremental Dose Intervals Improves Omics-Based Prediction for Acute Radiation Pneumonitis in Non-Small-Cell Lung Cancer Patients"

_cancers, 2022, doi:10.3390/cancers14194889_

Round 1

Reviewer 1 Report

Major limits

Results of this study are intriguing, but there still exist numerous limits, as described by the authors.

The selection of the patients is adequate, as they were all Stage III and treated with IMRT. Nonetheless, the fact that training set and testing set all derived from the same dataset (although the two groups were balanced) could hinder reproducibility and thus limit the impact of the study

Data derived from a retrospective single-institution study and this immnsely limit the reproducibility, that should be tested on multiple datasets from different Institutions.

Indeed, the main limit to the actual clinical implementation of radiomics is reproducibility (Salvestrini et al Transl Oncol. 2022 Jan;15(1):101275. doi: 10.1016/j.tranon.2021.101275.)

Try to better define the potential clinical impact and how the use of this model could modify radiotherapy and/or support treatment

Definition of comorbidities (especially pulmonary) could have been useful: is it possible to integrate such data?

Better clarify the feature selection progress

- line 240 ‘Of note, more than 50% of the selected dosiomics features came from lung subregions receiving less than 20 Gy … and approximately 20% for the selected radiomics features originated from lung sub-regions receiving less than 20 Gy’. Implement a potential explanation for this finding

Segmentation was performed on the basis of dose distribution. Could be there a role for anatomical segmentation as well? Was it performed?

Minor corrections

- line 20 simple summary: 'The image marker from radiomics and dosiomics features' should be changed to 'Markers derived from radiomics and dosiomics features...'

- line 30: 'and will be of great interest to anyone caring about radiation toxicity' should be changed into 'and could be useful to predict the risk of radiation-induced toxicity'

- abstract line 33: 'for predicting acute radiation pneumonitis (ARP) with grade ≥ 2 in lung cancer' should be changed into 'for predicting grade ≥ 2 acute radiation pneumonitis (ARP) in lung...'

- introduction line 61: 'ranking the second-top in its incidence rate and the top in mortality rate worldwide' should be changed into 'ranking at the second place in incidence rate and as the first in mortality rate worldwide'

- introduction line 64: 'severe ARP with grade ≥ 2' should be replaced with 'severe (grade ≥ 2) ARP' In all the subsequent text 'toxicity with grade' should be replaced with 'grade X toxicity' or equivalent periphrasis

- line 65 'enervate' should be replaced with 'hinder'

- line 69: I would remove examples of drugs, remarkably pirfenidone which is not currently widely approved for such indication

- line 70: 'with a hope to' should be replaced with 'with the aim to'

- line 80: 'emerge' should be replaced with 'advent'

- line 82: 'divulge' should be replaced with 'derive'

- line 110: 'analysis the responsiveness' replace with 'analyze the'

- line 138 patient data: 'physicians with more than' define the specialty of the physicians

- line 145: is it actually a radiologist that contoured the volumes? or is it a radiation oncologist? better define this issue

- line 158: 'SRs may provide a more thorough understanding on intra-pulmonary heterogeneity in radiosensitivity at a variety of dose levels'. Regarding this issue, you could also integrate the different anatomical radiosensitivity of lung segments, as defined in previous papers like Seppenwoolde et al. Int J Radiat Oncol Biol Phys. 2004 Nov 1;60(3):748-58. doi: 10.1016/j.ijrobp.2004.04.037.

- line 181: 'Since the order of 0, 0, 0 results in a constant value of 1, a total of 63 dose moments 181

were included in the dosiomics feature set' better clarify this example/definition

- line 213 'illuminated' should be replaced with 'summarized'

- line 235 'senior group': median age is actually quite low compared to other report and clinical practice, and the definition of 'elderly' in neoplastic patients and remarkably NSCLC is extremely debated (see Borghetti P et al Expert Rev Anticancer Ther. 2022 May;22(5):549-559. doi: 10.1080/14737140.2022.2069098. )

- line 236: 'belong to' should be replaced with 'affected by'

- line 290 discussion: 'in the long run. In this discussion, we attempted to highlight key findings of this study, scrutinize potential underlying reasons, and provide the community with recom- 292

mendations for future study' this part could be eliminated

- line 307: 'which suggests that severe ARP may present a stronger dependence on the radiomics features in the high-dose lung volumes' could be rephrased as 'which suggests a correlation of severe ARP and radiomics features in the high-dose lung volumes'

- line 336: 'Intriguing' should be replaced with 'intriguingly'

- line 352: 'This observation could be attributed to the complementary role between DF and RF for their contributions to the ARP occurrence' rephrase with '...for the identification of modifications underling ARP occurrence'

- The previous severe ARP modeling studies reported a wide range of performance with the

testing AUCs ranging from 0.68 to 0.94, as shown in Table S3. - Consider to include this table in the main text

- line 348: 'superiority of radiomics over dosiomics in divulging intrinsic tissue response upon treatment perturbation' provide adequate references and rephrase as '...in assessing intrinsic radiation-induced tissutal response'

Author Response

Dear Editor and Reviewers,

We sincerely thank the Editor and Reviewers for providing their feedback and thoughtful comments on our manuscript (Ref. No.: cancers-1823070). In the revised version, we have duly incorporated the feedback provided by the Reviewers and revised our manuscript by addressing comments to enhance the overall quality of our work. The manuscript has also been checked and corrected regarding typos and grammar errors. Comments from the reviewers are reproduced in black font followed by our responses in blue font. Major changes in the main manuscript are tracked in the revised manuscript, and summarized in this response letter. Thank you again for the time shared and consideration.

Yours Sincerely,

All Authors

September 16, 2022

Major limits

Results of this study are intriguing, but there still exist numerous limits, as described by the authors.

The selection of the patients is adequate, as they were all Stage III and treated with IMRT. Nonetheless, the fact that training set and testing set all derived from the same dataset (although the two groups were balanced) could hinder reproducibility and thus limit the impact of the study

Data derived from a retrospective single-institution study and this immnsely limit the reproducibility, that should be tested on multiple datasets from different Institutions.

Indeed, the main limit to the actual clinical implementation of radiomics is reproducibility (Salvestrini et al Transl Oncol. 2022 Jan;15(1):101275. doi: 10.1016/j.tranon.2021.101275.)

Response: Thanks for your professional comments. In the study, we only used data from our institution. In the study, we proposed a new subregion generation method (Incremental Dose-interval-based Lung Subregion Segmentation) by covering low-dose lung regions to improve the prediction for acute radiation pneumonitis. Results of this study would provide physicians with a more thorough understanding about contributions of different lung SRs at various dose levels in occurrence of ARP event in NSCLC patients. Recently, several publications also used single-center data and subregion method for predicting clinical outcome, for example Dong et al. Br J Radiol. 2022 Jan; 95(1129): 20201302. Doi: 10.1259/bjr.20201302 and Zhang et al. J Magn Reson Imaging. 2022 Jan; 55(6): 1636-1647. Doi: 10.1002/jmri.27993.

Meanwhile, we acknowledged multi-center validation which is essential and highly encouraged when it comes to putting the model into clinical practice. However, the key point of our study is focusing on a new subregion segmented method. The key results demonstrated that the IDLSS method improved model performance for classifying ARP with grade ≥ 2 when using dosiomics or combined radiomics-dosiomics features. Besides, we hope to provide an insight for researchers into optimizing study design in future works.

In the discussion (line 392-398), we already mentioned the limitation about the reproducibility. And, we also added a few of contents to highlight the limitation on reproducibility and cited the paper (lines 400-401) (Salvestrini et al Transl Oncol. 2022 Jan;15(1):101275. doi: 10.1016/j.tranon.2021.101275.)

Try to better define the potential clinical impact and how the use of this model could modify radiotherapy and/or support treatment

Response: Thanks for your professional comments. We modified the potential clinical impact (lines 116-120, 407-410). The description of “physicists in optimizing RT treatment plan in the long run” was removed from the whole manuscript. And the potential clinical impact of the study is mainly assisting physicians in prescribing timely interventions for at-risk patients.

Again, the key point of the study is focusing on the method comparison between our proposed dose-based lung segmentation (IDLSS) and the whole lung region, and the results demonstrated the potential power of our proposed method. As your previous mentioned, the common issue of radiomics model in the actual clinical implementation is reproducibility of the model. I think that a large cohort from multi-center data should be used for training a reproducible model in the future. At that time, our study could provide insight into design of future clinical studies that aim for patient stratification in a larger multi-center cohort.

Definition of comorbidities (especially pulmonary) could have been useful: is it possible to integrate such data?

Response: Thanks for your professional comments. In our dataset, the clinical information didn’t contain other comorbidities (especially pulmonary). In our study, we try to improve the prediction performance for ARP by means of our proposed dose-interval-based lung subregion segmentation (IDLSS) method. The IDLSS dose achieved a better prediction result for ARP comparing to the whole lung region. Therefore, a new study should be designed individually for integrating comorbidities (especially pulmonary).

Better clarify the feature selection progress

Response: Thank you for your comments. We modified the section of feature selection. The more details were given (lines 201-215). The modified part was shown below.

As shown in Figure 3 (b), feature selection was performed in an iterative process with five steps for each feature set. (1) At each iteration, 70% of the balanced data (88 cases) was subsampled independently from the whole data set. (2) For the sampled data, features with zero variances were filtered out, and the remaining features were standardized independently by removing the mean and scaling to unit variance (0-1) to avoid bias from data skew and scale [23]. (3) The student’s T-test was subsequently adopted to assess associations between the standardized features and the clinical outcome (ARP). Features with a p-value larger than 0.1 were further eliminated. (4) The previous iteration (steps of 1 to 3) was repeated over 100 times to obtain 100 feature sets. The most (10% of total feature number or at least top 10 features) frequently-occurring features were kept by considering the 100 feature sets and proceeded with feature redundancy test. (5) By calculating the Pearson correlation coefficient (R) between each pair of features, the feature with a higher mean correlation with the rest of the features in each highly-correlated feature pair (R > 0.5 [24]) was removed. Eventually, the remnant features were used for model construction.

- line 240 ‘Of note, more than 50% of the selected dosiomics features came from lung subregions receiving less than 20 Gy … and approximately 20% for the selected radiomics features originated from lung sub-regions receiving less than 20 Gy’. Implement a potential explanation for this finding

Response: Thanks for your comments. We added a few of contents in the Discussion (3rd paragraph, lines 314-324) following the advice:

The finding that over 50% of the selected dosiomics features originate from low dose subregions (<20 Gy) could be potentially ascribed to the larger inter-patient discrepancy in dose heterogeneity in low-dose regions than the high-dose regions. In IMRT treatment planning of lung cancer, highly conformal and homogeneous radiation dose deposition to the tumors (i.e., the high-dose subregion) are typically required for tumor control, this may likely lead to the phenomenon that the dose homogeneity within the high-dose subregions is less dissimilar between patients, and hence might not be sufficiently sensitive in predicting ARP. While in low-dose subregions, the inter-patient disparity in the dose homogeneity pattern of lung tissue is larger, which might to a degree explain the larger proportion of selected dosiomics features in these areas.

Segmentation was performed on the basis of dose distribution. Could be there a role for anatomical segmentation as well? Was it performed?

Response: Thanks for your professional comments. This is a great idea through considering anatomical segmentation. We didn’t perform this idea before. Generally, there are three ways for lung segmentation, including anatomy-based, lung function-based and dose-based. They all contribute to APR prediction to different extent. Our study belongs to dose-based method. To better demonstrated the method of our proposed IDLSS, we focused only on dose-partitioning. And we added some contents in introduction (lines 94-98) to specify our focused area.

Generally, lung segmentations in prediction of ARP can be divided into three categories, such as anatomy-based [13], lung function-based [14, 15] and dose-based [16–18]. The dose-based lung segmentation is widely investigated as considering the dose information into the radiation-induced toxicity.

We appreciated that the reviewer gave us a new scope for prediction on ARP using lung segmentation area, and thus promote our future work.

Minor corrections

- line 20 simple summary: 'The image marker from radiomics and dosiomics features' should be changed to 'Markers derived from radiomics and dosiomics features...'

Response: The sentence (lines 20-21) was modified by following the comments, as shown below:

“Markers derived from radiomics and dosiomics features can be adopted to predict ARP.”

- line 30: 'and will be of great interest to anyone caring about radiation toxicity' should be changed into 'and could be useful to predict the risk of radiation-induced toxicity'

Response: The sentence (lines 29-30) was modified by following the comments, as shown below:

The results could provide insight into the subregion’s ability in predicting the ARP, and could be useful to predict the risk of radiation-induced toxicity.

- abstract line 33: 'for predicting acute radiation pneumonitis (ARP) with grade ≥ 2 in lung cancer' should be changed into 'for predicting grade ≥ 2 acute radiation pneumonitis (ARP) in lung...'

Response: The sentence (lines 32-34) was modified by following the comments, as shown below:

“To evaluate the effectiveness of features obtained from our proposed incremental dose-interval-based lung subregion segmentation (IDLSS) for predicting grade ≥ 2 acute radiation pneumonitis (ARP) in lung cancer patients upon intensity-modulated radiotherapy (IMRT)”

- introduction line 61: 'ranking the second-top in its incidence rate and the top in mortality rate worldwide' should be changed into 'ranking at the second place in incidence rate and as the first in mortality rate worldwide'

Response: The sentence (lines 59-60) was modified by following the comments, as shown below:

“Lung cancer is one of the most common cancer malignancies, ranking at the second place in incidence rate and as the first in mortality rate worldwide”

- introduction line 64: 'severe ARP with grade ≥ 2' should be replaced with 'severe (grade ≥ 2) ARP' In all the subsequent text 'toxicity with grade' should be replaced with 'grade X toxicity' or equivalent periphrasis

- line 65 'enervate' should be replaced with 'hinder'

Response: The sentence (lines 63-64) was modified by following the comments, as shown below. Besides, we also modified the subsequent contents following the comments.

“Severe (grade ≥ 2) ARP can significantly degrade patients’ quality-of-life, cause treatment interruption, and thereby enervate therapeutic effect and worsen prognosis.”

- line 69: I would remove examples of drugs, remarkably pirfenidone which is not currently widely approved for such indication

Response: Following the comments, we remove the examples of drugs. Thanks for your professional suggestion.

- line 70: 'with a hope to' should be replaced with 'with the aim to'

Response: Following the comments, we modified the sentence (lines 68-70), as shown below:

Over the past decades, there have been extensive studies conducted with the aim to identify potential ARP risk factors, among which radiation dose has long been considered as the major attribute for inducing ARP in lung cancer patients

- line 80: 'emerge' should be replaced with 'advent'

- line 82: 'divulge' should be replaced with 'derive'

Response: Following the comments (lines 79-81), we modified the sentence, as shown below:

“More recently, the advent of Radiomics, enabling calculation of high-throughput imaging features from the entire tissue volume for cancer predictive modeling, has motivated scientists to derive three-dimensional spatial dose distribution information from dose-map images, referred to as Dosiomics.”

- line 110: 'analysis the responsiveness' replace with 'analyze the'

Response: Following the comments (lines 111-113), we modified the sentence, as shown below:

“analyze the responsiveness of each omics features to the whole lung region and lung subregions generating from IDLSS approach, which may provide a new prospective approach for radiomics related method.”

- line 138 patient data: 'physicians with more than' define the specialty of the physicians

Response: Thanks for your valuable comments. We specified the physicians as “radiation oncologist”. The sentence (lines 137-139) was modified as:

Radiation pneumonitis was graded following the Common Terminology Criteria for Adverse Events (CTCAE) V4.0 by radiation oncologist with more than 5 years of experience.

- line 145: is it actually a radiologist that contoured the volumes? or is it a radiation oncologist? better define this issue

Response: Thanks for professional comments. We replaced radiologist by “radiation oncologist” in the sentence (lines 145-147), as shown below:

The dose grid size was set as 3 mm for dose calculation, and the lung volumes were contoured by a radiation oncologist with more than 5 years of experience.

- line 158: 'SRs may provide a more thorough understanding on intra-pulmonary heterogeneity in radiosensitivity at a variety of dose levels'. Regarding this issue, you could also integrate the different anatomical radiosensitivity of lung segments, as defined in previous papers like Seppenwoolde et al. Int J Radiat Oncol Biol Phys. 2004 Nov 1;60(3):748-58. doi: 10.1016/j.ijrobp.2004.04.037.

Response: Thanks for your professional comments. We read the previous paper (Seppenwoolde et al.) carefully. They investigated the correlation between regional dose information and the radiation pneumonitis. The sub-regions were generated by the spatial position of lung that are the directions in anterior-posterior, caudal-cranial, distal-central, and contralateral-ipsilateral. In our study, subregional lung was segmented using incremental dose-interval-based method noted as IDLSS. Our proposed method is similar to the previous study, but they are totally different. We considered the subregion in a dose-interval based method, which is changed with the dose distribution. However, Seppenwoolde et al segmented lung as subregion using the lung’s spatial position. That subregion is much fixed and regardless with dose information.

Besides, we also added the information about the lung segmentation methods in the introduction (lines 93-96).

Generally, lung segmentations in prediction of ARP can be divided into three categories, such as anatomy-based [13], lung function-based [14, 15] and dose-based [16–18]. The dose-based lung segmentation is widely investigated as considering the dose information into the radiation-induced toxicity.

We are appreciated for reviewer’s suggestion and giving a new idea in anatomical radiosensitivity. Our team will discuss this new idea to further study the radiosensitivity and the correlation with the dose and anatomy.

- line 181: 'Since the order of 0, 0, 0 results in a constant value of 1, a total of 63 dose moments were included in the dosiomics feature set' better clarify this example/definition

Response: Thanks for your professional comments. We added a sentence to describe the definition of scale invariance with an equation. Based on the equation, we can simply calculate the scale invariance in the order of 0, 0, 0 as a constant value of 1. The added contents (lines 182-183) are shown below:

Scale invariance can be calculated by: , where ,  and  are the orders in three directions, and  is central moments which are defined in [19].

- line 213 'illuminated' should be replaced with 'summarized'

Response: Following the comments, we modified the sentence (lines 220-221), as shown below:

The flow chart of the model construction procedure used in this study is summarized in Figure 3 (a).

- line 235 'senior group': median age is actually quite low compared to other report and clinical practice, and the definition of 'elderly' in neoplastic patients and remarkably NSCLC is extremely debated (see Borghetti P et al Expert Rev Anticancer Ther. 2022 May;22(5):549-559.  . )

Response: Thanks for your professional comments. We modified the sentence following the previous paper. The modified sentence (lines 241-243) was shown below:

The average age of patients was 61 years-old with a standard deviation (STD) of 8.8 years, which is belong to the lower age group compared to other reports and clinical practice

- line 236: 'belong to' should be replaced with 'affected by'

Response: Following the comments, we modified the sentence (lines 243-244), as shown below:

As shown in Table 1, approximately two-thirds of the patients affected by squamous carcinoma cancer.

- line 290 discussion: 'in the long run. In this discussion, we attempted to highlight key findings of this study, scrutinize potential underlying reasons, and provide the community with recommendations for future study' this part could be eliminated

Response: Following the comments, we removed the part.

- line 307: 'which suggests that severe ARP may present a stronger dependence on the radiomics features in the high-dose lung volumes' could be rephrased as 'which suggests a correlation of severe ARP and radiomics features in the high-dose lung volumes'

Response: Following the comments, we modified the sentence (lines 322-325), as shown below:

In addition, approximately 50% (12 out of 22) of the selected RF stemmed from high dose regions of lung SRs (30-50Gy) (Table S1), which suggests a correlation of severe ARP and radiomics features in the high-dose lung volumes.

- line 336: 'Intriguing' should be replaced with 'intriguingly'

Response: Following the comments, we replaced “Intriguing” by “Intriguingly”.

- line 352: 'This observation could be attributed to the complementary role between DF and RF for their contributions to the ARP occurrence' rephrase with '...for the identification of modifications underling ARP occurrence'

Response: Following the comments, we modified the sentence (lines 370-371), as shown below:

This observation could be attributed to the complementary role between DF and RF for the identification of modifications underling ARP occurrence.

- The previous severe ARP modeling studies reported a wide range of performance with the

testing AUCs ranging from 0.68 to 0.94, as shown in Table S3. - Consider to include this table in the main text

Response: Following the suggestions, we moved the supplementary Table S3 in the main manuscript (Table 3) in the section of Discussion.  

- line 348: 'superiority of radiomics over dosiomics in divulging intrinsic tissue response upon treatment perturbation' provide adequate references and rephrase as '...in assessing intrinsic radiation-induced tissutal response'

Response: Following the comments, we modified the sentence (lines 366-367) as “This finding may probably be ascribed to the superiority of radiomics over dosiomics in assessing intrinsic radiation-induced tissutal response.”

Reviewer 2 Report

In this study, the authors compared the predictive models for radiation pneumonitis grade 2 or higher using radiomics and/or dosiomics features of whole lung and subregion of lung partitioned by radiation dose distribution, respectively.

The predictive models using radiomics and/or dosiomics features of subregion of the lung were superior to those using the features of the whole lung. The methods were well described, and 6 models using different features were appropriately compared. This study can provide some clues for developing machine learning models to predict radiation pneumonitis. I suggest this study is acceptable for publication in its present form.

Author Response

In this study, the authors compared the predictive models for radiation pneumonitis grade 2 or higher using radiomics and/or dosiomics features of whole lung and subregion of lung partitioned by radiation dose distribution, respectively.

The predictive models using radiomics and/or dosiomics features of subregion of the lung were superior to those using the features of the whole lung. The methods were well described, and 6 models using different features were appropriately compared. This study can provide some clues for developing machine learning models to predict radiation pneumonitis. I suggest this study is acceptable for publication in its present form.

Response: We sincerely thank the Editor and Reviewers for providing their feedback and thoughtful comments on our manuscript (Ref. No.: cancers-1823070). In the revised version, we have duly incorporated the feedback provided by the Reviewers and revised our manuscript by addressing comments to enhance the overall quality of our work. The manuscript has also been checked and corrected regarding typos and grammar errors.

Reviewer 3 Report

In this manuscript, the authors evaluated the effectiveness of an incremental Dose interval based lung sub region segmentation in the prediction of acute pulmonary pneumonitis following IMRT of non-small cell lung cancer patients. This is a well written and simply presented manuscript; Also, authors idea and conclusions are good and will help in guiding physicians in RT treatment plans.

Author Response

In this manuscript, the authors evaluated the effectiveness of an incremental Dose interval based lung sub region segmentation in the prediction of acute pulmonary pneumonitis following IMRT of non-small cell lung cancer patients. This is a well written and simply presented manuscript; Also, authors idea and conclusions are good and will help in guiding physicians in RT treatment plans.

Response: We sincerely thank the Editor and Reviewers for providing their feedback and thoughtful comments on our manuscript (Ref. No.: cancers-1823070). In the revised version, we have duly incorporated the feedback provided by the Reviewers and revised our manuscript by addressing comments to enhance the overall quality of our work. The manuscript has also been checked and corrected regarding typos and grammar errors.

Reviewer 4 Report

The article clearly discusses the results obtained and proposed a new IDLSS method that can support medical decisions.

Figure 3 is outside the margin of the sheet.

Author Response

The article clearly discusses the results obtained and proposed a new IDLSS method that can support medical decisions.

Figure 3 is outside the margin of the sheet.

Response: We sincerely thank the Editor and Reviewers for providing their feedback and thoughtful comments on our manuscript (Ref. No.: cancers-1823070). In the revised version, we have duly incorporated the feedback provided by the Reviewers and revised our manuscript by addressing comments to enhance the overall quality of our work. The manuscript has also been checked and corrected regarding typos and grammar errors. Besides, we modified the format of Figure 3 following your comments.

Round 2

Reviewer 1 Report

The paper has been appropiately edited and the revised version is suitable for publication